# Colostrum Composition, Characteristics and Management for Buffalo Calves: A Review

**DOI:** 10.3390/vetsci10050358

**Published:** 2023-05-18

**Authors:** Daria Lotito, Eleonora Pacifico, Sara Matuozzo, Nadia Musco, Piera Iommelli, Fabio Zicarelli, Raffaella Tudisco, Federico Infascelli, Pietro Lombardi

**Affiliations:** Department of Veterinary Medicine and Animal Production, University of Napoli Federico II, 80100 Napoli, Italy; daria.lotito@unina.it (D.L.); pacificoeleonora7465@gmail.com (E.P.); saramatuozzo98@gmail.com (S.M.); fabiozicarelli@gmail.com (F.Z.); tudisco@unina.it (R.T.); infascel@unina.it (F.I.); pietro.lombardi@unina.it (P.L.)

**Keywords:** colostrum, buffalo, cow, colostrum management, colostrum quality, buffalo farm, buffalo breeding

## Abstract

**Simple Summary:**

Buffalo breeding has high economic, cultural, and historic importance around the world. In Italy, most of the farms are located in the Campania region with 58.4% of herds and 66.6% of animals mainly milked for the production of “Buffalo Mozzarella Cheese”. The neonatal period is very critical for buffalo calves and is often characterized by a high mortality rate, therefore, it is fundamental to ensure optimal passive immunity within the first few days of life. For these reasons, the intake and absorption of colostral immunoglobulins are critical to attain passive protection against infectious diseases. As a consequence, the management and administration of colostrum are also critical since its characteristics need to be preserved to guarantee its efficacy.

**Abstract:**

In this review, the composition, characteristics, and management of dairy buffalo calves were examined and compared with bovines. The neonatal period is critical for buffalo calves and is characterized by a high mortality rate (more than 40%). The early intake of high-quality colostrum (IgG > 50 mg/mL) is the one way to improve the immune system of calves (serum IgG > 10 mg/mL after 12 h), thus increasing their chances of survival. Mainly in intensive farms, the availability of high-quality colostrum is necessary; thus, good quality colostrum is often stored to provide newborn calves which cannot be fed by their mothers. Also, the manipulation of the immunological status of animals through vaccination has been depicted since the quality of colostrum tended to be influenced by vaccination against pathogens. Buffalo breeding is constantly expanding in Italy, mainly thanks to the Mozzarella cheese production that represents the excellence of the “Made in Italy” and is exported worldwide. Indeed, high calf mortality rates directly affect the profitability of the business. For these reasons, the aim of this review was to examine specific research on buffalo colostrum that, compared with other species, are scarce. Improving the knowledge of buffalo colostrum, in terms of characteristics and management, is critical to guarantee buffalo newborns’ health in order to reduce their mortality rate. Importantly, considering the knowledge on cattle valid also for buffalo is a widespread, and often erroneous, habit in several fields, including colostrum feeding. Therefore, the two species were compared in this review.

## 1. Introduction

Buffalo newborns are more susceptible to infections than those of other animal species, because the maternal Ig is not transported across the placental membrane and neonates lack Ig, thus, colostrum intake plays a decisive role in passive immunity since calves need at least two weeks to start producing their own antibodies [1]. In general, despite some studies that have been performed to compare buffalo and bovine physiology [2,3] research on buffalo is limited. Concerning colostrum, compared to cow [4], sheep [5] and goat [6,7], research on buffalo is scarce, mainly regarding its quality. In this review, the bovine colostrum has been considered for comparison with the buffalo one. High calf mortality rates directly affect the profitability of farms, whether the herd is destined for meat or milk production. Proper feeding, in terms of quality, quantity, and frequency of administration, of colostrum to the newborn calf is crucial, several studies have shown that incorrect feeding and or incorrect colostrum management do not allow the calf to form an efficient immunity. The National Database of the Zootechnical Registry statistics, from 2017 to 2022, showed an average mortality rate of 41.11% during the first six months of life of buffalo calves [8].

Buffalo breeding has high economic, cultural, and historic importance mainly in developing or developed countries. In Italy, most of the farms are located in the Campania region with 58.4% of herds and 66.6% of animals [9] mainly milked for the production of “Buffalo Mozzarella Cheese” [10]. This is an example of how quality and tradition represent identity, but also income and employment. The system along the certified supply chain generates a direct turnover of almost 600 million euros with a remarkable effect on the local economy. “Mozzarella di Bufala Campana DOP” is, after the “Grana Padano”, “Parmigiano Reggiano” and “Gorgonzola”, the fourth Italian DOP in terms of quantity, with an annual production of over 54,000 tons (2021), 35% of which is exported (France and Germany alone account for more than half of the quantities exported) [11]. In addition, in recent years, the interest in buffalo meat is also increasing in Italy, due to its nutritional characteristics [12,13,14].

Buffaloes, similar to all ruminant species, have the syndesmochorial placenta which prevents the passage of antibodies from the mother to the newborn, thus, calves are very vulnerable at birth being agammaglobulinemic or hypogammaglobulinemic with little to no transplacental passage of immunoglobulins (mainly IgG) [15]. In the first days of life, the immune system needs to be guaranteed by the absorption of colostral immunoglobulins, to ensure passive protection against infectious diseases (diarrhea, respiratory diseases, etc.) and for the future health and growth of the calf [16]. During pregnancy, the passage of nutrients to the fetus occurs by the placentomes, consisting of cotyledons, portions of the placenta, and caruncles, portions of the uterine horns. Placentomes do not allow the passage of antibodies: this is the reason why colostrum intake plays a crucial role for the newborn [17].

Colostrogenic transformation starts many weeks prior to calving when immunoglobulins are transported from the serum to the mammary gland [18]. Late in the pregnancy, this process continues, but it ends abruptly when the calf is born. Mammary gland secretory epithelial receptors allow the transfer of immunoglobulins from maternal blood to the mammary gland under the influence of lactogenic hormones and regional regulatory factors [19]. Ruminant colostrogenesis begins in the last weeks of pregnancy and ends immediately post-calving [20]. The blood cells and other colostral components migrate to the mammary gland [21,22], where their phenotype and function are altered [23,24]. During colostrogenesis, the transfer of cellular and non-cellular components from the maternal circulation into colostrum occurs in the mammary gland. The transfer of immunoglobulins (Ig) from maternal circulation to mammary secretions, begins 3–4 weeks before parturition under endocrine control [25]. It is essential that the newborn calf receives the correct amount of colostrum, a natural source of macro and micro-nutrients [26] and peptides with antimicrobial activity and growth factors [27]. The concentration of immunoglobulins represents the main factor to assess the quality of colostrum. The permeability of the gut decreases over the first 24 h after calving, so, in order to achieve a good immunization, the calf needs to ingest colostrum within 4–6 h of age [28]. Squillacioti et al. [29] focused this issue on aquaporins, which are membrane channel proteins that regulate water permeability in many tissues [30], presuming that their manifestation in the gut of buffalo calves is affected by a proper ingestion of colostrum after delivery.

As seen, buffalo breeding has reached high economic importance in Italy and worldwide, nevertheless, despite a very high neonatal mortality rate that has been recently reported specific research on buffalo colostrum is still scarce. Thus, it appears that improving the knowledge of buffalo colostrum, in terms of characteristics and management, may be important to give practitioners and farmers useful information to guarantee buffalo newborns’ health. This is the principal aim of this review. Importantly, considering the knowledge on cattle valid also for buffalo is a widespread, and often erroneous, habit in several fields, including colostrum feeding. For this reason, the two species were compared in this review.

## 2. Colostrum

### 2.1. Characteristics and Composition

To favor healthy growth, newborn calves need nutrients like fat, proteins, vitamins, minerals, lactoferrin, immune cells, cytokines, and immunoglobulins (Ig) [31]. Calves assimilate colostrum nutrients during the first hours of life [32]. Changes in colostrum composition depend on many factors, such as the time passed from calving, parity (i.e., the primiparae have different Ig content), age of the animal, dry period feeding, and length of the dry period of cows [33]. Differences in management and nutrient concentration between small and large farms [34], as well as in the composition of colostrum between geographic zones (high and low altitude) [35] have been also reported. In Table 1 and Table 2 the mean chemical composition of buffalo and bovine colostrum is reported.

Colostrum contains 30–200 g/L proteins [44] mainly represented by antibodies (IgG1, IgG2, IgA, IgM) [45,46]. Fat is the most variable constituent [47]. El-Fattah et al. [36] showed that the concentration of total protein and whey protein, composed of β-lactoglobulin, α-lactalbumin, proteose peptone, serum albumin, and immunoglobulins in that sequence of abundance [48] did not differ between buffalo and cow colostrum, while fat and total solids were significantly higher in buffalo colostrum. Another factor affecting colostrum composition in both buffaloes and cows is the breeding season, mainly concerning fat content, fatty acids profile, and cholesterol [49,50]. The total protein and Ig content in the first colostrum is higher in buffalo cows than in dairy crossbred cows [51].

The high protein content is mainly due to the large amount of immunoglobulin [52]. Bioactive compounds in colostrum can be directly observable through laboratory analysis, as is the case for immunoglobulin G1 [53], growth hormone (GH), prolactin (PRL), insulin, and glucagon [54]. Colostrum contains insulin-like growth factors (IGF-1 and IGF-2) in high quantities and can improve the development and function of the gastrointestinal tract of newborn calves, and stimulate tissue, body growth, and development in newborn calves [55,56].

Comparing the buffalo and bovine colostrum, some differences emerged, particularly concerning fat and protein content. The fat content appears higher in buffalo (ranging from 9.59 to 18.75%) compared to bovine, in which the lipid content ranged from 4.60 to 6.70%. On the contrary, the protein values showed an opposite trend (ranging from 5.44 up to 13.46% in buffalo vs. 12.4 up to 14.9% in cows) and there is less variability among the studies available in the literature. Similar values for lactose, total solids, and ash were observed. Colostrum has also an important immunoregulating function and modifies the gut microflora, thanks to the presence of the bifidogenic glycoproteins in colostral whey demonstrating the immunoprotective role of the colostrum glycoconjugates. [57]. Antigenic glycopeptides and macrophage-activating oligosaccharides [58], similar to those found in bovine and human colostrum, were identified in buffalo colostrum [59]. Ashok et al. [60] focused on whey proteins from buffalo colostrum and analyzed the effect of peptides derived from peptic-digestion on DNP-induced oxidative stress on blood components. It may be inferred that the combination of empirical and bioinformatic approaches facilitated the unraveling of various functionalities of early milk peptides that contribute to protecting and improving neonatal health during the early part of its growth and development [61].

The minerals in ruminant colostrum (Ca, Mg, P, and Na) decrease gradually from the fifth day after delivery. The intensive mineral requirement of the newborn calf may account for the high level of ash content in the colostrum [62]. Colostrum has also a thermoregulation function for the survival of neonatal ruminants [63]. Colostrum energy content may affect the thermoregulation and fatty acid oxidation necessary to sustain gluconeogenesis [64]. Newborn calves, fed on higher volumes of colostrum, exhibited increased thermoregulatory responses, improved growth performance, and higher immunity [65]. Another function of colostrum is the laxative one, as it allows the calf to start forming its immunity by contributing to the proper expulsion of meconium [66]. In Table 3 the principal colostrum functions are reported.

### 2.2. Colostrum Quality

Differences among species influence the quality of colostrum as well as parity, animal health, and farm management. The first 24–48 h after calving are representative of the colostrum quality in terms of IgG concentration [67]. After 24 h, the gut no longer absorbs immunoglobulins, which remain active in the intestinal lumen acting as a local defense mechanism against agents that may have been ingested orally. To determine colostrum quality, several markers have been proposed (i.e., crude protein, total protein, gamma-glutamyltransferase), the same parameters can be assayed in calf blood to evaluate the extent of the passive immunity transfer [68]. Comparing the buffalo with cow colostrum showed remarkable differences. El Fattah et al. [36] described that, at calving, the protein concentration does not differ between buffalo and cow colostrum, while total solids, fat, lactose, and ash concentrations were higher in buffalo than in cow colostrum. Also, all components decreased gradually as the transition period advanced except lactose, which conversely increased. Godden et al. [69] and Pempek et al. [70] reported that the survival rate and health of calves were correlated with the improvement of serum IgG in colostrum. In addition, other key factors contributing to this list are the quantity and speed of administration of colostrum, and no less significant, its cleanliness and quality.

On the contrary, in a study by Toro-Mujica et al. [71], a relationship between the amount of colostrum supplied and calf mortality was not observed.

### 2.3. Colostrum Analysis

The bovine colostrum quality can be estimated by the colostrometer, a device that measures the specific gravity related to the total gamma-globulin concentration and enables an estimation of quantity on the basis of a statistical correlation. In the buffalo, a non-significant correlation between the specific gravity and total gamma-globulin was seen [72]. The reason lies in the different colostrum composition between cow and buffalo colostrum, with higher concentrations of fat and proteins in buffalo colostrum that may affect the colostrometer readings. Within all parameters evaluated as possible markers for buffalo colostral quality, the enzyme GGT appeared with the highest correlation observed between its activity and the concentrations of gamma globulins [73,74].

The enzyme gamma-glutamyltransferase is a cell membrane-bound enzyme located on the outer surface of cell membranes. It transports and uses glutathione (GSH), a tripeptide with antioxidant action, as a source of amino acids for cell metabolism [74]. Results of Pero et al. [74], showed that glutathione is secreted in buffalo colostrum and that the enzyme GGT uses it as a substrate for its activity.

Younger buffalo cows have lower immunoglobulin content in the colostrum than older ones [75]. This is related to the increased antigenic exposure in older cows, so that a greater array of antibodies is transferred from bovine serum to the colostrum. Moreover, the mammary gland also plays an important role, and, since its development may not be complete in the younger cows, the transport of IgG may be reduced [40] According to Pritchett et al. [76], animals at the second lactation showed significantly lower IgG concentration than those at the third lactation. In general, the number of bio-active components in colostrum decreases after 6/12 h from birth. In addition to its immunological importance, the colostrum has also been reported to possess anti-inflammatory [77], anticancer [78], antimicrobial [79], and nutraceutical [80] properties.

Compared to bovine colostrum, buffalo colostrum shows higher concentrations of lactose, ash, total solids, fat, vitamin E, phosphorus, and IGF-1, and lower of vitamin A, Mg, K, Na, Zn, and lactoferrin [36]. Colostrum contains other bioactive compounds, such as lactoferrin (LF), lysozyme, and growth factors [39], that will be discussed later.

Colostrum pasteurization, used to prevent pathogens contamination, could cause undesirable changes in nutritional, functional, and physicochemical properties [81,82] mainly with temperatures above 60 °C, which lead to the denaturation of bioactive proteins [83,84,85] Colostrum composition is affected by pasteurization either for long (at 63 °C for 30 min) or short period (at 72 °C for 15 s) [86,87,88] In order to preserve bioactivity and to improve the quality and shelf life of cow and buffalo colostrum, the combination of light pasteurization (at 57 °C for 30 min) and freeze-drying should be used. Indeed, buffalo colostrum shows higher bioactivity than that of bovine [39]. The measurement of the IgG concentration makes it possible to evaluate the quality and to monitor colostrum feeding practices. The first assessment is visual: the colostrum appears yellow and very thick and creamy. The yellow color is due to the presence of maternal antibodies [62].

Afterward, assessment tools are needed, such as a refractometer or a colostrometer, that show the content of IgG. The colostrometer measures specific gravity and, using a color scale calibrated in milligrams per milliliter (mg/mL) of immunoglobulins (Ig), it converts specific gravity to Ig concentration. Generally, colostrum that tests “green” contains > 50 mg/mL of Ig, “yellow” contains 20 to 50 mg/mL, and “red” contains < 20 mg/mL of Ig [62]. The use of the colostrometer is a commonly practiced procedure, but it suffers from poor accuracy and wide variability with a relatively low correlation to the concentration of immunoglobulins. On the contrary, the refractometer is more accurate, and it presents a Brix scale. The refractive index of the serum estimates the protein concentration, which is an indication of the concentration of immunoglobulins [89]. The Brix value of 22% corresponds to 50 mg/mL, and with such value, colostrum can be considered of high quality. Several studies explored the utility of the Brix refractometer for monitoring colostrum management programs and guaranteeing passive immunity transfer (PIT) in bovines. The digital Brix refractometer seems to be an affordable tool for buffalo farms, allowing to use of the same device for monitoring both colostrum quality and the success of passive transfer in calf serum. Indeed, relatively few studies have been performed, additional research, mainly evaluating a high number of animals, would be necessary to set specific cut point values [90,91]. Also, most studies were carried out only to evaluate the buffalo’s colostrum quality, the thermal stability, and some physicochemical characteristics [92]. In any event, Brix is a good screening parameter for management purposes to see the colostrum quality because there is a correlation between refractive index and IgG concentration.

## 3. Passive Immunity Transfer (PIT)

The calf must acquire passive immunity via colostrum [47]. The timing of colostrum intake, the method, and volume of colostrum administration, the presence of the mother, as well as respiratory acidosis of the newly born, are linked to its absorption in calves [93]. Colostrum antibodies can protect the calf for up to six weeks. During this period, in the environment, the animal contacts infectious agents, which gradually stimulate the development of its immune system [28]. If the calf does not receive colostral immunoglobulins, is highly exposed to diseases, and this can even cause a high mortality rate of calves on the farm [94,95]. The onset of immunocompetence (active immunity) occurs after three weeks of age, when the passive coverage of the antibodies taken up via colostrum begins to decrease [96].

Feeding colostrum, immediately after birth, is crucial for the calf’s health because the highest calf’s ability to absorb immunoglobulins from colostrum decreases as early as six hours after birth [97]. During the dry period, two months before delivery, the tight junctions of the mammary gland epithelium are not yet perfectly developed so that the epithelium is still loose [98]. This allows macro-molecules (soluble immune factors, anti- bodies, and blood cells) that are present in the mother’s blood to pass between the secreting cells arriving directly in the alveolar lumen. This also occurs for antibodies during fetal life when the animal is preparing for birth. During that period, a direct passage of antibodies into the alveolar lumen can occur, since the newborn calf, during the first two days of life, has not yet fully developed the intestinal tight junctions, which allow the absorption of macromolecules through the intestinal wall [99,100]. With regard to the immunocompetence of calves, Barmaiya et al. [101] deduced that the amounts of IgM, IgG, and IgA in serum samples from buffalo calves are lower than the amounts from cow calves. In cows, IgG, IgA, and IgM account for approximately 85% to 90%, 5%, and 7%, respectively, of the total Ig in colostrum [102,103].

El Fattah et al. [36] reported similar values of immunoglobulins comparing buffalo and cow colostrum. In detail, the concentrations of IgG and IgM were 33.20 and 3.00 mg/mL in buffalo and 32.33 and 3.20 mg/mL in cow colostrum, respectively, and in both species, they significantly decreased after five days from parturition.

In Table 4 and Table 5, the concentration of Ig, respectively in buffalo and bovine colostrum, is reported. In both species, as depicted in Table 4 and Table 5, there was a high variability concerning the Ig content that ranged from 30 up to 86% of IgG in buffalo and from 55 up to 90% in cow colostrum. Lower values were registered for IgA in both species, but the buffalo showed higher variability compared to the cow (from 0.18 up to 23% for buffalo and 1.66 up to 7% for cow colostrum, respectively). Instead, the IgM showed more constant values in both species. It is important to underline that in the cited papers the time of colostrum sampling could be different, and this aspect can be decisive in determining the Ig content in colostrum.

The absorption capacity of immunoglobulins in the buffalo calf can be determined by quantitatively assessing the concentrations in blood [108]. In large ruminants, the absorption efficiency of immunoglobulins decreases over time, in particular, if administered 6 h after birth [109], about 66% are recorded in the plasma, after 12 h about 50%, and so on [110,111]. In buffalo, Dang et al. [112] observed the highest IgG activity in colostrum samples collected on day one after calving, and a significant decrease was observed by day seven.

Research has shown that colostrum quality can be improved by supplementing vitamins before birth. The impact of antioxidant vitamins on mineral and Ig secretion in colostrum was studied [113], showing that, thanks to some vitamin supplementation, the buffalo cow voluntarily ingests more feed before delivery, causing an increase in weight and, consequently, also the colostrum is of higher quality [69,111]. Moeini et al. [114] showed that Se and Vit. E administration in heifers improves significantly (*p* < 0.05) their concentration in colostrum.

Few specific studies have been conducted in buffalo on this topic, El-Loly et al. reported that IgG has important effects against enteric and respiratory infectious diseases [105]. Interestingly, a recent study showed that buffalo colostrum has useful therapeutic proteins that can also be used in humans [79]. With regard to intestinal absorption of colostrum in calves, Pero et al. reported [115], that CaSR (Ca^2+^- sensing receptor) and Na+K+ATPase play a critical role in identifying the intestinal characteristics that enable the absorption of the colostrum nutrients.

Infascelli et al. [116] showed that integrating aloe into buffalo feed improves passive immune transfer in newborn buffalo calves, as it increases the immunological properties of the colostrum. In particular, colostrum from mothers supplemented with aloe during their last months of pregnancy showed an increase in colostrum IgG content. Results indicated that supplementing aloe to the dry ration of mothers can improve colostrum immunological qualities, which in turn improves passive transfer in newborn calves. Furthermore, it has been observed that the supplementation of vitamins A and E has a positive influence on the colostrum composition of Murrah buffaloes [117]. In general, supplementing dams is considered a good strategy to enhance colostrum quality, An et al. [118] reported a positive effect of prepartum maternal *Capsicum oleoresin* supplementation in buffalo. In particular, these authors demonstrated that it improved calves’ growth performance and increased weekly starter intake.

## 4. Colostrum Management and the Impact on Calves

### 4.1. Colostrum Storage

Colostrum production rapidly ceases during parturition, and its composition drastically varies with time, making it necessary to collect it from suitable donors as soon as possible [115]. Pathogenic bacteria and pH are significantly altered during the storage and changes are more rapid when colostrum is stored at temperatures > 4 °C [119]. There are studies that sustain that colostrum can be kept in plastic containers at room temperature, but the shelf life is just three days due to deterioration and fermentation [41,120]. It has also been studied that the viability of cellular components and immunoglobulins, at a temperature of 4 °C, keeps them stable for up to a week [121].

Freezing high-quality colostrum to make it available when necessary is a common practice in buffalo farms. Colostrum should be collected to refill the colostrum bank with new supplies. It is advised to give colostrum right away after it has thawed. After freezing, the content of immunoglobulins does not change, but it needs attention in the phase of defrosting [91]; for example, high temperatures and incorrect agitation can compromise the composition. In contrast, El-Fattah et al. [122] found similar levels of IgG, IgM, IGF-1, and lactoferrin in freeze-drying bovine and buffalo colostrum compared with fresh ones. These findings may be due to the freeze-drying preserves heat-sensitive biological components, since the processing temperatures are low and there is a rapid local transition of the frozen material from the hydrated to dehydrated state, which minimizes the denaturation of protein.

Colostrum pasteurization is not typically advised to decrease colostrum bacterial numbers as it is linked to immunoglobulin degradation and consistency issues [92]. El-Fattah et al. [122] comparing the effect of pasteurization of bovine vs. buffalo colostrum at different temperatures and times. In particular, at 63 °C for 30 min or 72 °C for 15 s they found significantly lower IgG, IgM, IGF-1, and lactoferrin concentrations (*p* < 0.05) than those in raw colostrum and this result may be due to the denaturation of colostral immunoglobulins by heating via an initial reversible unfolding of native structure, with loss of globular configuration. Instead, pasteurization at 60 °C for 60 min caused no significant difference in colostral IgG and viscosity, while a significant (*p* < 0.05) reduction was observed in IgM, IGF-1, and lactoferrin concentrations of buffalo colostrum as compared with raw buffalo one. Salar et al. [38] studied the differences between pasteurization and drying processes on cow and buffalo colostrum. This research showed that, by increasing the time and temperature of pasteurization, the IgG concentration in both species significantly (*p* < 0.05) decreased. In particular, a reduction of IgG was observed changing the temperature from 57 to 60 (around 5.8%) and up to 63 °C (around 13%) in cow colostrum, whereas in buffalo the mean loss was 9.3 and 10.3%, respectively. Buffalo colostrum seems more sensitive to heat treatment than cow one. Also, the lactoferrin content decreased by 57 and 51% in cow and buffalo, respectively, with an increase from 60 up to 63 °C of pasteurization. The way colostrum is administered should be taken into account because it might affect the timing of the first feeding, the amount ingested, and the effectiveness of Ig absorption [123]. Colostrum should be kept chilled to prevent bacterial growth if it is not fed within two hours from collection. So, after quality evaluation, and if it exceeds 3–4 L, colostrum is stored in freezers, in thermal bags where the buffalo number and the correspondent brix can be identified. The differences observed among the cited bibliography suggest that the chemical composition and type of colostrum could affect its shelf life.

### 4.2. Benefits of Colostrum: Infection Prevention

Both buffalo and bovine calves are extremely vulnerable; thus, they are exposed to a high risk of infection. Guidelines recommend feeding 10 percent of the calf’s weight of colostrum to the bovine calf [46].

Enteropathogens, such as bovine rotavirus, bovine coronavirus, *Escherichia coli,* and Cryptosporidium are the main culprits causing calf diarrhea during the animal’s first 30 days of age [124]. Regarding *Escherichia coli*, calves are most vulnerable to contracting Enterotoxigenic *Escherichia coli* infection, which causes watery diarrhoea. Following ingestion, *Escherichia coli* colonizes the gut by adhering to the enterocytes (intestinal epithelial cells) of the intestinal villi and infects the gut epithelium. Due to the low pH, the distal part of the small intestine offers the best conditions for *Escherichia coli* colonization (less than 6.5), so it infected the small intestine and causes secretory diarrhoea in calves [125].

The intake of colostrum prevents contamination by viruses, parasites, and bacteria [126]. This is also due to the presence of lysozyme, which provides the calf with many benefits. Lysozyme is an enzyme with antibacterial action produced by macrophages; its action focuses on the destruction of peptidoglycans that build the cell wall of gram-positive bacteria. The ingestion of colostrum, which contains high amounts of lysozyme, can contribute to the prevention of neonatal infections also in buffalo calves [127,128]. Enteropathogens are the main culprits causing calf diarrhea during the animal’s first 30 days [129]. For example, rotaviruses are present in the environment, and they are very contagious. Adult animals are the main source of infection for young calves, especially between 1 and 3 weeks. The virus is fecally transmitted, and calves are often infected by contact with other calves primarily or secondarily through objects, feed, and water. In addition, rotaviruses have a wide host range, infecting many animal and human species including buffalo [123,130,131].

Nevertheless, specific research on buffalo is still poor. *Toxocara vitulorum*, a parasitosis that mainly affects buffalo and cattle calves [131] by localizing in the small intestine causes obstructive phenomena that can result in failure to absorb nutrients from the feed and subsequent decreased growth, diarrhoeic phenomena and, in some cases, death of the calf [132]. Trans-colostral and trans-milk transmission is also recorded for this species, in fact, in buffaloes, the larvae become encysted at the muscle level becoming hypobiotic larvae that, only shortly before parturition, will reactivate by migrating into the mammary gland and colostrum [132]. It is also possible to increase the specific resistance of newborn buffalo calves against Pasteurella Multocida by vaccinating the dam before parturition in order to promote the production of specific antibodies [133].

### 4.3. The Effects of Vaccination on Colostrum Quality

The ability to manipulate the immunological status of animals through vaccination against diseases that affect humans and the opportunity to harvest those immunoglobulins in the form of colostrum or milk is a topic of interest in both animals and humans. Immunization against major pathogens, such as rotavirus and coronavirus, the most common cause of diarrhea, can enhance the overall passive transfer of immunity to the offspring by the colostrum administration. In the research of Menichetti et al. [134] a significant increase (*p* < 0.05) in colostral IgG (around 19%) in cow vaccinated at 28 dpp compared to cow not vaccinated are reported. Saif et al. [135] showed that calves that did not receive colostrum from cows vaccinated against rotavirus exhibited diarrhea. The quality of colostrum tended to be influenced by vaccination against pathogens. Ali et al. [136] showed that the colostrum of buffalo dams that underwent vaccination had a richer content of IgG and a higher percentage of total solids, solids-not-fat, total protein, fat, and lactose. Such colostrum components were the highest at the birth time, then they decreased gradually up to 72 h after birth except for the percentage of fat and lactose which showed gradual increases up to 72 h to reach the normal composition of milk. In a review proposed by Maunsell et al. [137] the vaccination improved the colostrum quality if compared to unvaccinated to vaccinated cows at 3/6 weeks prior to calving. In particular, IgG is transferred to the newborns via colostrum and are fundamental to preventing many diseases in the early period of life. In the study conducted by Jayappa et al. [138], it was shown that vaccinating cows early in pregnancy, particularly at 3 months, can provide passive protection to newborn calves through antibodies transferred via colostrum, against a etiological agents such as *E. coli* [138]. Denholm et al. [139] and Denholm et al. [140] reported that colostrum samples for vaccinated herds had a higher quality than samples for unvaccinated ones. Souza et al. [141] evaluated the passive immunity transfer in healthy buffalo calves and found significant improvement in colostrum quality in vaccinated herds. Civra et al. [142] showed that the conventional bovine rotavirus vaccine was able to boost the anti-HRoV protective efficacy of bovine colostrum suggesting a conservative, feasible, and not yield-limiting approach to produce hyperimmune colostrum that could represent a functional feed to prevent and treat HRoV infections. Vaccination of pregnant cows significantly reduced calf morbidity and mortality rates [143]. Buffalo calves born to immunized dams exhibit high levels of rinderpest-neutralizing antibodies thanks to the intake of colostrum [144].

As reported by Hulbert et al. [145] and Svensson et al. [146] the action of providing adequate amounts of high-quality colostrum in the first six hours after birth is critical in reducing the severity of health problems and, consequently, the number of calf deaths.

## 5. Conclusions

Colostrum plays a fundamental role in the survival of buffalo calves. The purpose of this review was to analyze the benefits of passive immunity transfer provided to calves through colostrum administration and the benefits of the latter. Thus, we provided an updated picture of the specific research carried out on buffalo colostrum. Although the differences with the bovine species have been repeatedly underlined by various authors, the management of colostrum is based, often incorrectly or superficially, on information deriving from research carried out on bovines. For this reason, where possible, the differences in the literature between the two species as far as colostrum is concerned have been underlined. In addition to improving management aspects, current studies on buffalo aim to improve the quality of colostrum, developing methods to strengthen the immune system of the calf, such as vaccination of mothers close to delivery, or the administration of supplements in pregnant buffaloes in order to improve the colostrum quality.

## Figures and Tables

**Table 1 vetsci-10-00358-t001:** Composition of Water Buffalo colostrum (mean ± standard deviation).

Nutrient					
	El -Fattah et al. [36]	Yonis et al. [37]	Coroian et al. [25]	Salar et al. [38]	Bernabucci et al. [39]
Fat (%)	9.59	9.70 ± 0.50	11.31 ± 0.39	10.81 ± 0.50	18.75
Protein (%)	13.46	11.93 ± 0.55	8.73 ± 0.15	7.97 ± 0.76	5.44
Lactose (%)	.	2.50 ± 0.60	3.73 ± 0.02	3.78 ± 0.06	2.7
Total Solids (%)	26.67	25.20 ± 0.60	25.31 ± 0.02	23.93 ± 1.46	.
Ash (%)	.	1.10 ± 0.05	0.94 ± 0.02	0.97 ± 0.02	.
pH	.	.	6.01 ± 0.01	6.04 ± 0.02	.

**Table 2 vetsci-10-00358-t002:** Composition of bovine colostrum (mean ± standard deviation).

Nutrient						
	Dunn et al. [40]	Salar et al. [38]	Godden et al. [41]	Kehoe et al. [34]	Elfstrand et al. [42]	Arslan et al. [43]
Fat (%)	6.40	6.11 ± 0.78	6.70	6.70	4.60	6.40
Protein (%)	14.0	12.91 ± 3.75	14.0	14.92	12.4	14.0
Lactose (%)	2.70	3.21 ± 0.41	2.70	2.49	3.0	2.70
Total Solids (%)	.	23.33 ± 4.11	23.90	27.64	.	.
Ash (%)	.	1.01 ± 0.08	1.11	0.50	.	0.50
pH	.	6.26 ± 0.09	.	.	.	.

**Table 3 vetsci-10-00358-t003:** Colostrum functions in newborn calves.

Colostrum Functions in Newborn Calves
Passive Immunity Transfer [27,34,41,64]
Nutritional function in terms of fat, protein, and vitamin [25,31,36,38,42,43]
Thermoregulation [40,63]
Development of the gastrointestinal tract [55,56]
Body growth improvement [55,56]

**Table 4 vetsci-10-00358-t004:** Total Ig concentration in buffalo colostrum.

Immunoglobulin	Dang et al. [104]	El -Fattah et al. [36]	El-Loly et al. [105]	Bernabucci et al. [39]	Goel et al., Elfstrand et al. [42,106]
IgG (%)	86	33.20	74.46	54.0	30–36
IgA (%)	8	.	22.89	3.22	0.18–0.57
IgM (%)	6	3.00	2.65	5.22	0.47–0.57

**Table 5 vetsci-10-00358-t005:** Total Ig concentration in bovine colostrum.

Immunoglobulin	Stelwagen et al. [27]	Godden et al. [41]	Costa et al. [107]	Arslan et al. [43]
IgG (%)	86	85–90	88.39	55.00
IgA (%)	7.0	7.0	4.25	1.66
IgM (%)	7.0	5.0	4.71	4.32

## Data Availability

The data presented in this review are available on request from the corresponding author.

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
