# Peer review of "Colostrum Composition, Characteristics and Management for Buffalo Calves: A Review"

_vetsci, 2023, doi:10.3390/vetsci10050358_

Round 1

Reviewer 1 Report (New Reviewer)

Journal: Veterinary Sciences
Manuscript ID: vetsci-2363344
Type of manuscript: Review
Title: Colostrum composition, characteristics and management for buffalo
calves: a review 
Authors: Daria Lotito, Eleonora Pacifico, Sara Matuozzo, Nadia Musco *, Piera
Iommelli *, Fabio Zicarelli, Raffaella Tudisco, Federico Infascelli, Pietro
Lombardi
Submitted to section: Veterinary Physiology, Pharmacology, and Toxicology

The authors aim to provide the current view for the composition, characteristics and management for dairy buffalo calves and compared with bovines. The management and administration of colostrum is also critical since its characteristics need to be preserved to guarantee its efficacy. This review is really interesting and obtain the significant knowledge how calves can enhance their performance, high mortality rate or passive immunity. I consider that the review could be suggest to publication after minor revision and please see comment below.

 Abstract:

-Suggest adding some essential data into this section.  

-At the end of section, suggest adding the summarized of present review.

Introduction:

-Suggest extending this section in order to show what is the significant problem of the current situation and why the author require to review.

-L41-42: Please provide reason why?

-L45-47: Why current feeding did not “Proper feeding”? Please explain more.

-Table 1,2,4,5: Please check reference format eg El -Fattah et al. 2012…it should be El -Fattah et al. [34]…check for all.

-L112: Please classify “antibodies”

L134-135: Please provide biological mechanism why Colostrum also has an important immunoregulating function and modifies the gut microflora??

-Table 3: Change to “Passive immunity transfer”

-L196-197: Please provide biological mechanism why “Younger buffalo cows have lower immunoglobulin content in the colostrum than the older ones”

-L253: What is “Feeding colostrum at the right time”? Please describe.

L287-189: Please provide biological mechanism?

L294: Table 4 should be change to Figure 4?? Also, please define superscript a to d.

Conclusion:

This section should be focus answer the objective of the review and reply to the significant of the problem. The key conclusion is not well clear.

-L446-449: Recommend removing since it seem duplicate and not suitable place for detail of objective.

Reference:

-Formatting regarding to Journal guidline.  

Minor editing of English language required

Author Response

please find attached the replies to reviewer 1 

Reviewer 2 Report (New Reviewer)

The subject of the article is interesting and valuable, showing practical value of improving quality of the colustrum.

But overall manuscript is quite erratic, repetitions occuring regularly - so in the reviewer's opinion manuscript still needs editorial work of the Authors.

There is lack of analysis of hygienc aspects of milking for colostrum collection, as well as health issues of cows influencing on colostrum quality. Not all active substances presented in colostrum were described.

The reviewer would expect from the Authors practical tips for improving quality of colostrum, as well as underlying the needs of education of farmers.

Please check the 1st sentence added to the abstract, lines 19-20 - "colostrum" word is missing.

The whole abstract should be re-written, to avoid duplications of the content and focus more on the presentation of clues of the article.

Itroduction part 1. is very limited. Reviewer suggests to add separate chapter for points 1.1 & 1.2. Part 1.2 is quite erratic.

line s207 & 208 LTLT and HTST pastuerization formulas

Conclusions part is too long.

Some of the references are quite old. 

The overall quality of the language is fine, giving the fluency of reading. Minor grammar mistakes found. 

Author Response

please find attached the replies to reviewer 2

Reviewer 3 Report (New Reviewer)

Comments and Suggestions for Authors

The authors of this review examined and compared the composition, characteristics and management for dairy buffalo calves with bovines. The authors demonstrated that considering the knowledge on cattle valid also for buffalo is a widespread, and often erroneous, habit in several fields, including colostrum feeding by comparison. They have shown the importance of colostrum in the first hours of calves’ life.

The review was balanced with a sufficient number of references. The methods of analysis are good. The manuscript was well written. 

Line 121: There's an extra comma after [51]

Line 286: Dang et al. 2007 [112] and in all table; line 294-296: El-Loly et al. reported that IgG has important effects against enteric and respiratory infectious diseases [105]; line 298: Pero et al. reported [114]; line 301: Infascelli et al. [115] showed that integrating aloe into buffalo feed improves passive… The way authors are cited should be reviewed.

Line 391: Put the scientific name ''Pasteurella Multocida'' in italics, e.g. Pasteurella Multocida”.

Author Response

please find attached the replies to reviewer 3

Round 2

Reviewer 1 Report (New Reviewer)

Thank you very much for agreeing to the revision, both my suggestion and the current amended version are more appealing to the reader and should be published as is.

Reviewer 2 Report (New Reviewer)

The reviewer sees some improvement of the manuscript. Most of previous suggestions were considered by the Authors.

This manuscript is a resubmission of an earlier submission. The following is a list of the peer review reports and author responses from that submission.

Round 1

Reviewer 1 Report

The main element missing from the manuscript is the effect of buffalo vaccination on colostrum quality in the postpartum period. This is a key element whether the quality of the colostrum is better or not with the correct vaccination schedule of the animals. In a way, this can be related to the size of the animals and why older animals have better colostrum quality. That would mean they've been in contact with more pathogens.

To what extent and what parameters of buffalo colostrum compared to the parameters and properties of bovine colostrum are affected by its preservation processes?

A table/graph comparing buffalo and bovine colostrum would be a valuable element

Author Response

Please find attached the revisions.

Reviewer 2 Report

The article is poorly structured and the title misleading.  This is not a review of colostrum management for buffalo calves but a general article about colostrum.

Many of the references are wrongly quoted- table 2- reference seems to refer to protein concentration in water buffalo, Checking the reference (93) it refers to bovines.

Reference 100 is from work in cattle yet is quoted as a buffalo reference. 

Extensive work on this paper is required before consideration for resubmission.

Author Response

Please find attached the revisions.

Reviewer 3 Report

Dear Authors 

I appreciate your attempt to review this important topic "colostrum management". I recommend reconsideration of this manuscript after major revision due to the following reasons. After significant modifications, this manuscript can be submitted for reviewing again.

1. If this manuscript have line numbers it would have been very easy to mention changes.

2. It is suggested to mention "Mozzarella di Bufala Campana DOP" in english also. As its difficult to understand for readers of other countries and languages.

3. It is suggested to add 2-3 more keywords.

4. Why buffalo newborns are susceptible to infection, mention in first line of the introduction.

5. In table 1, compare composition of buffalo colostrum mentions by different researchers to make this manuscript more informative.

6. Page 3, last paragraph, it is suggested to add a figure showing the different/important functions of colostrum.

7. Combine section 4 with 2.3.

8.  In table 2 also, mention and compare the immunoglobulin composition by researchers in recent studies. researchers have explored it after 1982 also.

9. Include latest data in table 3 also.

10. It is suggested to add significant data and studies in section 5, as the manuscript is written on this topic only "Colostrum management".

11. After strengthening section 5, conclusion and abstract need to be updated accordingly.

I recommend submission of manuscript after making suggested changes.

Author Response

Please find attached the revisions.

Round 2

Reviewer 1 Report

There is a big improvement in the manuscript.

However, it is still not enough for a review of buffalo colostrum alone. As other reviewers have also suggested - it's best to reword the text and change the title. Make a detailed comparison between buffalo and bovine colostrum. It will carry more value and it will definitely be more interesting. The more so that the literature on this subject is quite poor, and the authors rely a bit exaggeratedly on literature that does not fully indicate buffalo colostrum as one of the reviewers pointed out.

Author Response

please find attached the replies.

Reviewer 3 Report

The manuscript quality has been improved significantly after revision. For further improvements, following suggestions need to be incorporated.

    1.   In table 2, add sources/references to the colostrum function.

    2. Section 4.2 needs to be included under section 2.1.

    3. As there is only one section “4. Colostrum management and the impact on calves” for colostrum management in this manuscript. Whereas, whole manuscript present significant information on composition and characteristics. Thus, it is suggested to modify manuscript title to “Colostrum composition, characteristics and its management for Dairy Calves”.

Author Response

please find attached the replies.
